# Tree rings reveal globally coherent signature of cosmogenic radiocarbon events in 774 and 993 CE

Ulf Büntgen et al.[#]

Though tree-ring chronologies are annually resolved, their dating has never been independently validated at the global scale. Moreover, it is unknown if atmospheric radiocarbon enrichment events of cosmogenic origin leave spatiotemporally consistent fingerprints. Here we measure the $^{14}$C content in 484 individual tree rings formed in the periods 770–780 and 990–1000 CE. Distinct $^{14}$C excursions starting in the boreal summer of 774 and the boreal spring of 993 ensure the precise dating of 44 tree-ring records from five continents. We also identify a meridional decline of 11-year mean atmospheric radiocarbon concentrations across both hemispheres. Corroborated by historical eye-witness accounts of red auroras, our results suggest a global exposure to strong solar proton radiation. To improve understanding of the return frequency and intensity of past cosmic events, which is particularly important for assessing the potential threat of space weather on our society, further annually resolved $^{14}$C measurements are needed.

The precise dating of high-precision proxy archives is essential for climate reconstructions, as well as for their comparison with climate forcing agents, climate model simulations, historical sources and archaeological evidence. Extra-tropical wood formation usually generates distinct annual stem growth increments. Consecutive patterns of tree-ring width therefore allow continuous and often well-replicated records of multi-centennial to millennial length to be developed if suitable measurements from living trees are successfully cross-dated against relict wood from historical timbers, archaeological excavations and/or subfossil remains[1]. The annual dating accuracy of such composite chronologies is not only crucial for (paleo-) climatology and ecology[1,2], but also for advancing interdisciplinary research that engages archaeology, history and the environmental sciences[3]. Although intercontinental assessments reveal a high level of growth coherency between individual ring width chronologies from different parts of the world[2], their dating precision has never been independently corroborated at the global scale. Even a 1-year dating offset would prevent any straightforward comparison of tree-growth anomalies with climate forcing agents, output from climate model simulations, climate-induced environmental responses or documented socio-economic and political changes. A prime example of such interdisciplinary cross-comparison is the cluster of large volcanic eruptions identified from ice-core anomalies in 536, 540 and 547 Common Era (CE), which coincided with rapid cooling at the onset of the Late Antique Little Ice Age (LALIA)[3], as well as the outbreak of the Justinian Plague in 541 CE and large-scale societal transformation and human migration across much of Eurasia.

If of sufficient amplitude, abrupt changes in the Earth's atmospheric radiocarbon ($^{14}C$) abundance[4,5] are recorded in tree rings owing to the short mixing time of the atmosphere[6], including stratosphere-troposphere exchanges. Often attributed to extreme fluxes of high-energy solar particles[7,8], distinct $^{14}C$ anomalies in 774/5 and 992-4 CE[4,5,9-13], as well as possibly much earlier in 660 and 3372/1 BCE[14,15] have been identified in local proxy archives. These so-called Cosmic Events also yield anomalies in records of other cosmogenic radionuclides, such as $^{10}Be$ and $^{36}Cl$ that are measured in ice cores[7]. Though proposed as a "paradigm for chronology"[16], the extra-terrestrial origin, spatial variation and time-transgressive evolution of cosmogenic tie-points are still unknown because a spatially extensive, well-replicated network of isotopic markers across continents and hemispheres has not hitherto been available.

In an unprecedented voluntary collaboration of the international tree-ring community, the COSMIC initiative gathered wood samples from the majority of the world's longest tree-ring records, spanning the decades 770–780 and 990–1000 CE. The resulting COSMIC network includes data from 44 disjunct tree-ring width measurement series (Supplementary Note 1), with sampling locations on five continents (Fig. 1; Supplementary Fig. 1). With the individual sites being distributed between 42° S and 72° N, and ranging from around 40 to 4000 m a.s.l., COSMIC represents most of the world's extra-tropical forest biomes. Covering four atmospheric radiocarbon zones[17], our dataset is composed of 13 genera, including 25 coniferous and two broad-leaf species (Supplementary Tables 1–2). Wood fibres from each tree ring, dendrochronologically cross-dated to the period 770–780 CE, or, if not reaching that far back in time, to the interval 990–1000 CE, were manually separated, chemically extracted and automatically graphitized (Methods; Supplementary Table 3). Modern compact tandem accelerator mass spectrometry (AMS) requires 1000-times less material and operates as precisely as some traditional decay counters[18] (Methods). Developed and based at ETH-Zurich, the "Mini Radiocarbon Dating System" (MICADAS)[18] was used to measure the $^{14}C$ content of 30–50 mg bulk cellulose for a total of 484 tree rings. A subset of 374 rings in the 770s CE interval originates from 27 records on the Northern Hemisphere (NH) and seven records on the Southern Hemisphere (SH). Another 110 rings that did not reach back into the 8th century CE represent eight NH and two SH records in the 990s CE.

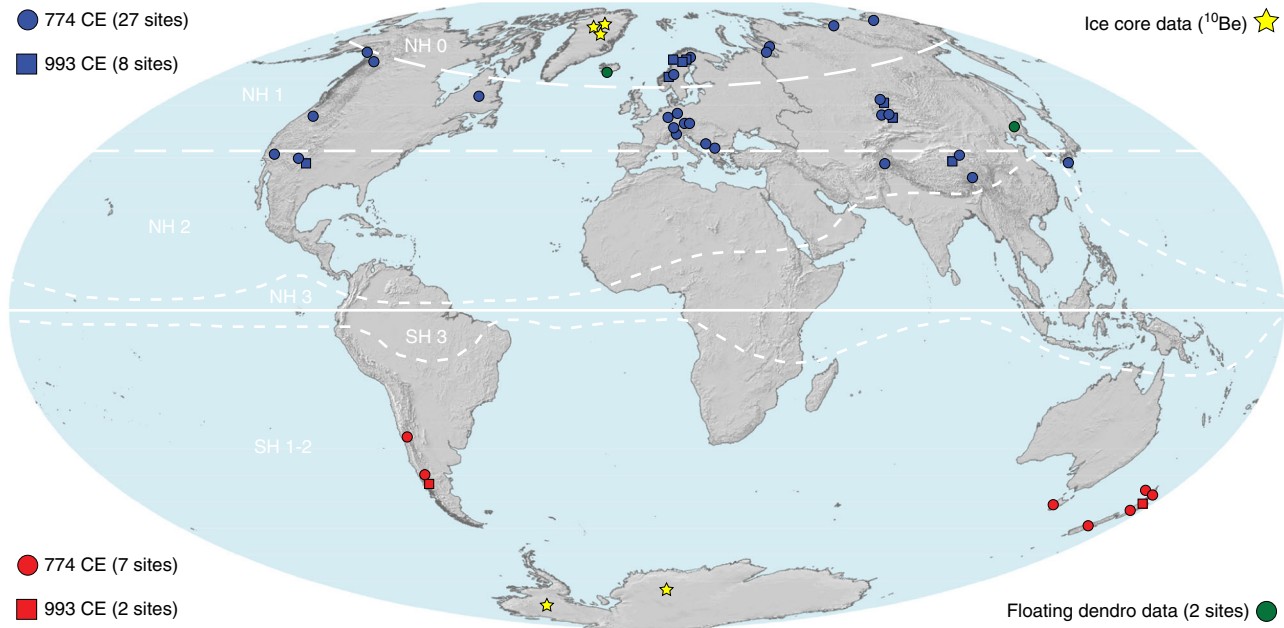

**Fig. 1** COSMIC network. Distribution of 44 tree-ring records from which cellulose was extracted for annual $^{14}C$ measurements during the intervals 770–780 and 990–1000 CE (circles and rectangles) (Supplementary Fig. 1; Supplementary Tables 1-2). Independent $^{14}C$ evidence from two floating tree-ring chronologies[21, 22] (green), and five (quasi) annually resolved ice-core $^{10}Be$ records[23, 24] (yellow). White dashed lines refer to atmospheric radiocarbon zones[17]. The map reflects knowledge from the authors and was created via software ArcGIS 10.1 SP1 for Desktop by Esri

## Results

**Cosmogenic radiocarbon signature in 774 CE**. The coherent, rapid increase of $^{14}$C concentration in NH and SH tree rings reveals the global extent of the cosmogenic signature of extreme fluxes of high-energy solar energetic particles (SEP) in 774 (Fig. 2a). The lower $\Delta^{14}$C values in the SH most likely reflect different ocean circulation, wind and land–ocean–atmosphere exchanges[6,19]. For the NH, the individual $^{14}$C site values, as well as their corresponding hemispheric range, show low levels in 773 ($\Delta\Delta^{14}$C = −0.1%), higher concentrations in 774 (0.4%), a further increase in 775 (0.95%), a relatively small increase in 776 (0.2%), and decreasing concentrations thereafter. Although the overall shape of the $^{14}$C time-series through the event is similar for both hemispheres, its amplitude is lower in the SH, and, in contrast to the NH signal, the $^{14}$C increase in 774 (0.7%) is comparable to the increase in 775 (0.6%) (Fig. 2a). Collating results from both hemispheres into four meridional bands reveals a similar time-transgressive pattern (Supplementary Fig. 2). The tree ring-based $^{14}$C concentrations predominantly reflect tropospheric conditions during wood formation, rather than concentrations of radiocarbon in the stratosphere where it is mostly produced. We therefore employ a global carbon box model to generate mechanistic understanding of the intra-annual course of the radiocarbon cycle (Supplementary Fig. 3). With 95% confidence, the model times the event, presumed to have been ephemeral, within the boreal growing season (June to August) of 774 CE (Supplementary Fig. 4a; Supplementary Table 4). Importantly, this seasonal timing is consistent with the observed ~10% relative difference in radiocarbon amplitude between the NH and SH.

**Cosmogenic radiocarbon signature in 993 CE**. Similar in shape to the 8th century event is a significant, but smaller $^{14}$C excursion in 993 (Fig. 2b). The best model/data agreement at the 95% confidence interval suggests it occurred between February and June 993 (Supplementary Fig. 4b; Supplementary Table 4). In contrast to the 8th century excursion, the sharp radiocarbon enrichment in 993, found in both hemispheres, is followed by a moderate increase from 993 to 994 CE. This behaviour is indicative of a late boreal spring event. Although associated with uncertainties, our monthly resolved model assumes that about 2/3 of the total annual wood biomass is produced within 1–2 months in the middle of the vegetation period (Supplementary Fig. 3). Since this rather short window of cell formation and cell wall thickening is synchronized between all sites in each hemisphere, phenological changes in growing season length have no influence on the model outcome. Moreover, only a very small impact is expected from so-called carry-over effects, because <10% of stored carbohydrates from previous year(s) is typically used for cell growth. The model also accounts for seasonal differences in stratosphere–troposphere $^{14}$C exchange, and thus simulates the occurrence of both radiocarbon enrichment spikes in the boreal summer (July ± 1 month) of 774 and the boreal spring (April ± 2 months) of 993 (Supplementary Fig. 4) with great confidence. The intra-annual timing of the $^{14}$C event in 993 CE translates into a coherent signature across both hemispheres. While the global extent and time-transgressive evolution of the 774 and 993 events are similar, the amplitude of the latter event is about half that of the former. With our box model, an additional 5.3 ± 0.5 (9.6 ± 0.5) $10^{26}$ atoms, i.e. 1.8 ± 0.2 (3.2 ± 0.2) times the annual production, are needed to produce the 993 (774) anomaly (Supplementary Fig. 4). Although our findings demonstrate accurate cross-dating of well-replicated tree-ring chronologies from around the world, they still cannot guarantee the annual precision of individual measurements from single trees (Methods). The independent replication of tree-ring width measurement series therefore remains an essential element in dendrochronology, which further distinguishes our discipline from most other proxy archives for which sample size is much more restricted.

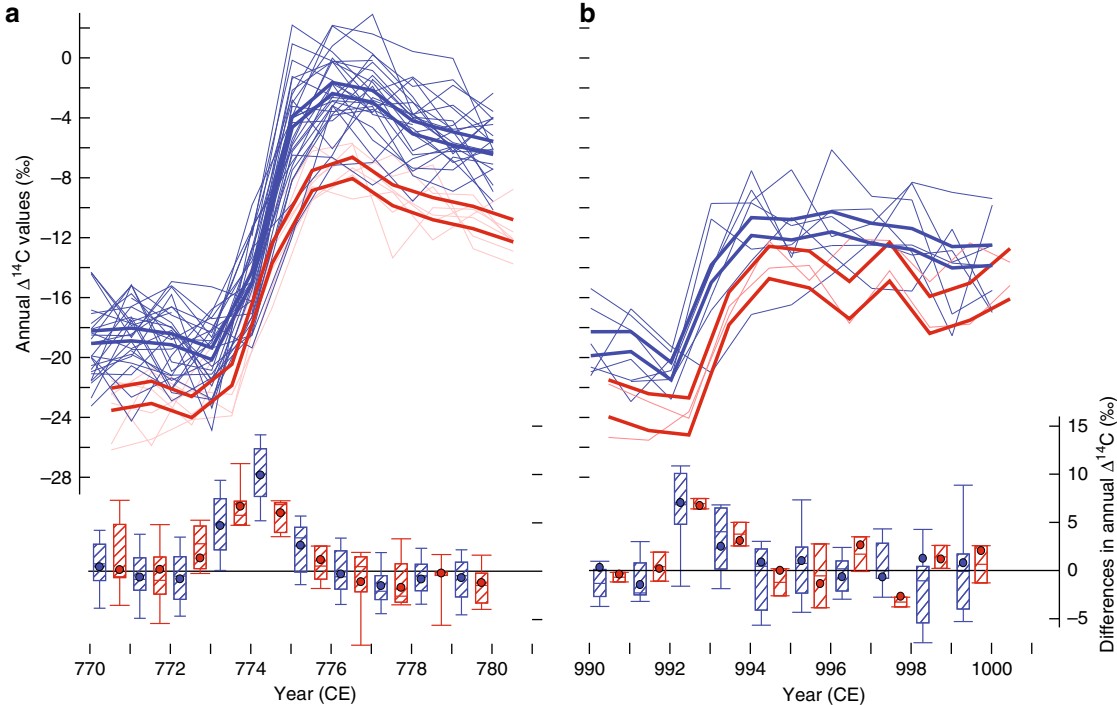

**Fig. 2** COSMIC signature. **a** Annual $^{14}$C content of 374 tree rings formed between 770 and 780 CE at 27 and seven sites across the NH and SH (light blue and red lines), respectively (Supplementary Fig. 2). **b** $^{14}$C content of 110 tree rings from 990–1000 CE at eight and two sites in the NH and SH (blue and red lines), respectively. Thick lines bracket standard uncertainties around hemispheric means, lower box plots reveal year-to-year $^{14}$C differences (median, 25th and 75th percentiles), and SH data have been shifted relative to the earlier NH data

## Discussion

Following the LALIA, the boreal summer of 774 over most of the NH was ~0.7 °C colder than the 1961–90 reference period[20], whereas 993 coincided with medieval summer warmth of ~0.6 °C (relative to the 1961–90 mean climatology). Although speculative, slightly lower mean temperatures in the 770 s (compared to the 990 s) may have contributed to a larger, though insignificant (4.0‰ ± 0.4 in 774 versus 3.5‰ ± 0.7 in 993), hemispheric offset in the radiocarbon concentration (Fig. 2). In contrast to previous, model-based assumptions[19], we hypothesize that an overall warmer climate in the 990 s might explain the smaller interhemispheric $^{14}C$ difference during high medieval times via reduced atmospheric mixing and/or ocean upwelling in the SH. Our limited understanding of how, if at all, climate change affects the amount of outgassing $^{14}C$-depleted $CO_2$ from the proportionally larger SH oceans, however, calls for more research. Moreover, different ecological site conditions and species-specific plant physiological processes, including xylogenesis, may further impact the timing and intensity of cellulose-based $^{14}C$ anomalies through varying rates of cell formation and carbon sequestration[12]. Such influences are expected to be particularly strong in multibiome, global scale tree-ring compilations[2], such as the COSMIC network, which contains wood from living trees, historical timbers, archaeological excavations and subfossil remains of 27 species.

Of greater significance than the ~3–4‰ $^{14}C$ inter-hemispheric offset (Fig. 2), we identify a meridional north–south gradient of declining average $^{14}C$ values in each 11-year tree-ring record (Fig. 3). Independently of the two cosmogenic events in the 770s and 990s, the mean atmospheric radiocarbon concentration is generally higher at northern latitudes >60° N. The $^{14}C$ transport to lower latitudes where the growing season of trees is usually lengthening, however, might be altered by stratosphere–troposphere exchanges[6]. Complex ocean circulations, climate gradients and different growing season lengths are expected to alter further the large-scale distribution of the Earth's radiocarbon zones. Since the high $^{14}C$ values at the most northern sites cannot be explained by these processes alone, further research into the role of ocean extent and winds[6,19], ocean–atmosphere circulation and coupling, terrestrial and solar magnetism, and the combined effects on the global $^{14}C$ dispersal, is needed.

The global signature of the two cosmogenic excursions independently validates the precise dating of well-replicated tree-ring

chronologies from five continents. Moreover, it offers a unique opportunity for assessing and/or correcting records within and between proxy archives, akin to the nuclear Bomb Peak of the 1960s (Methods). This study further emphasizes the accuracy and relevance of using high-resolution $^{14}C$ measurements in determining the age of floating tree-ring series that have been used to date the eruptions of Katla (Iceland)[21] and Changbaishan (China and DPR Korea)[22] and (Fig. 1). Quasi-independent evidence of the two radiometric time markers in the 770 and 990 s is also found in high-resolution $^{10}Be$ deviations in five ice cores from Greenland and Antarctica[23,24] (Fig. 1; Supplementary Table 5).

Given the exceptional nature of these two SEP events, we have examined contemporary medieval texts to see if any references might attest to these cosmic events. We recognize that interpretation of such documents is contested and acknowledge the potential for misinterpreting these narratives. The prominent Japanese Buddhist priest Kûkai (Kôbô Dashi), born in 774, was given the official name Henshô Knonô, often translated as "worldwide shining diamond"[25]. Moreover, "... a person from India presented himself before the Caliph al-Mansur in the year 776 who was well versed in the *siddhānta* method of calculation related to the movement of the heavenly bodies, ..."[26], and in the same year there was a gathering of Mayan mathematicians and astronomers at Copán in Honduras[27]. Together with other writers, Britton[28] refers to an aurora when talking about a red cross that appeared in the heavens after sunset between 773 and 776. Again, the 18th century physician Thomas Short (1690–1772) repeated a record from *Chronicon Magdeburgensis*[29]: "993 On the 7th of the Calends of January, at one a Clock in the Night, suddenly Light shined out of the Night like mid-day; it lasted an Hour, but the Sky turning red, the Night returned.". Some references have been interpreted as aurora[28–30], but may, alternatively, suggest point-of-light events, such as a gamma ray burst or supernova. Our data, however, suggest globally homogeneous impacts in 774 and 993 that can be best explained by large energy releases from the Sun[7], such as SEP events. Historical records from Germany, Ireland and the Korean Peninsula suggest the occurrence of red auroras between late-992 and early-993 CE[31], which could be interpreted as great magnetic storms from intense solar activity. While preceding previously reported $^{14}C$ results from local analyses by one year[5,14], the medieval observations are consistent with our findings.

Together with advances in accelerator mass spectrometry and the availability of continuous, multi-millennial tree-ring chronologies from different parts of the world, our results highlight the importance and feasibility of reproducing the decadal-resolved IntCal calibration curve at annual resolution over much of the Holocene and maybe even beyond[32]. Such an improved high-resolution radiocarbon record would provide unprecedented dating confidence to various fields of the natural sciences and humanities. It would also bring a deeper understanding of the frequency, magnitude and origin of cosmic events, critical to assessments of the threat of space weather to our society[33].

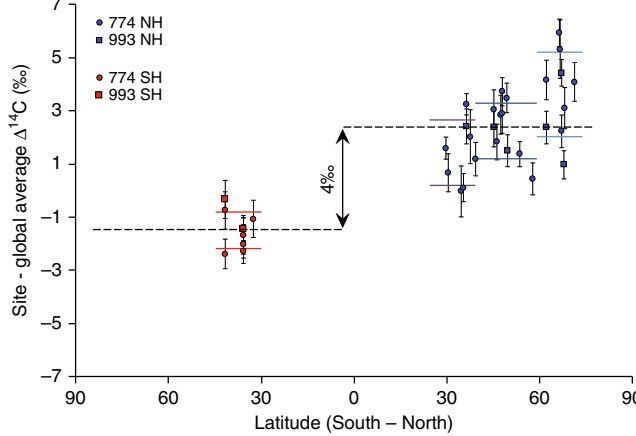

**Fig. 3** COSMIC gradient. The $^{14}C$ content of individual tree-ring site records averaged over 11-year intervals from 770–780 and 990–1000 CE. Horizontal lines indicate atmospheric radiocarbon zones[17]. Grey lines are standard errors of the tree-ring site records. Linear trends within the NH and across both hemispheres are $R^2 = 0.29$ and 0.71, respectively (not shown)

## Methods

**Sample preparation**. Although the highest level of caution has been applied in each step throughout the COSMIC project, the final $^{14}C$ measurements may still contain some degree of uncertainty, because of a wider range of possible biases and errors inherent to such a community wide, voluntary collaboration (Supplementary Note 1).

Once identified, the original data provider, usually an experienced tree-ring researcher either at a university, institute or laboratory, be it private or governmental, delivered the right wood sample that comprises either the 770–780 or 990–1000 interval CE. Although this sounds trivial as individual samples are usually coded after sampling and before archiving, the relevant material often originates from fieldwork campaigns that, in some cases, have been conducted several decades ago. It is important to note, that both samples and scholars may have changed location and workplace throughout the years. A systematic, long-

term archiving of wood samples from either extremely long-lived trees, historical timbers, archaeological excavations or subfossil remains, is often complicated because many universities, institutes and laboratories, smaller or larger, simply do not have the appropriate storage capacity. A sustainable archiving of relict wood samples, especially in the case of wet archaeological and subfossil material, poses a serious challenge, and funding for appropriate facilities of such long-term "service" is generally scarce.

Once the right sample was identified and found in one of the many COSMIC-relevant archives around the world (Supplementary Table 1), an accurate, high-resolution re-measurement of the individual tree-ring widths of the two decades of interest (770–780 and 990–1000 CE) was performed to ensure precise cross-dating. Each original data provider was asked to mark the rings in question, sometimes smaller than 0.1 mm if trees were growing at or near their species-specific distribution limit (see above). The labelled (coded) and marked (stitched) wood samples were then packed and sent to the Swiss Federal Research Institute WSL in Birmensdorf (Switzerland), where all material was quality controlled (e.g. wood composition), and further wood preparation steps were performed (see details below).

At WSL, each individual ring boundary from 770–780 or 990–1000 CE was re-identified to guarantee the correct ring width sequence within the two 11-year intervals. All rings were then manually cut with a scalpel and the corresponding wood material was carefully divided and fragmented into similarly sized fibre flakes. Even the smallest contamination either from unclean tools or neighbouring older and younger latewood and early cells, in cases of less defined or irregular ring boundaries, can cause a substantial bias. Contamination is particularly critical when running the high-resolution $^{14}C$ measurements with the MICADAS[18], simply because the total amount of material is only 20–50 mg (see below). Wood samples, ideally representing the interior part of the ring to avoid boundary contamination and seasonal differences in wood formation, were transported to the Laboratory of Ion Beam Physics at ETH Zurich (Switzerland), where all radiocarbon measurements were performed.

**Radiocarbon measurements**. Wood fibres of each tree ring (Supplementary Tables 1-2), putatively dendrochronologically cross-dated to either the 770–780 or 990–1000 CE interval, were manually separated, chemically extracted and automatically graphitized (Supplementary Table 3). All fibres were soaked in NaOH over night at 70 °C, before being washed for 1 h with HCl and again 1 h in NaOH. All fibres were finally bleached in 5% $NaClO_4$ for 2–3 h at 70 °C until they turned white[12]. A total of 2–3 mg of cellulose per individual ring sample was wrapped in tin capsules and graphitized in the automated graphitization line "AGE"[34,35] (see details below). The cellulose-based graphite was then analysed for its radiocarbon concentration with the high-precision compact accelerator mass spectrometer MICADAS[18]. This modern AMS was developed and is located at ETH Zurich. Subdivided into 10 cycles of 30 s, total measurement time was 1–2 h/sample, with an OX-II standard typically yielding ~500,000 counts or more ($^{12}C$-ion currents of 50 μA) (see details below). Radiocarbon-dead kauri (*Agathis australis*) wood from New Zealand and brown coal from Germany[12], as well as an absolutely dated historical reference timber from 1515 CE[1], were considered for $^{14}C$ normalization and independent quality control.

The raw wood from each annual tree-ring sample was chemically extracted to holocellulose following a modified version of the BABAB protocol[36], i.e. the base-acid-base-acid bleaching method for holocellulose extraction before graphitization[37]. Sequential steps in the procedure are shown in Supplementary Table 3. Samples were washed at near neutral pH with de-ionized water between each processing step. The combination of sample combustion with an elemental analyser and the subsequent graphitization of the $CO_2$ with $H_2$ on iron powder is widely used to prepare graphite targets for radiocarbon measurement by AMS. Graphitization via "AGE"[34,35] considers pre-cleaned iron as a catalyst, for which iron was first baked for three minutes in ambient air, and then reduced under $H_2$ atmosphere (at 80 kPa and 500 °C for 5 min) twice, followed by a third reduction step (80 kPa $H_2$, 500 °C, 20 min). The reaction vessels of the AGE consist of Duran glass tubes that are prebaked (500 °C, 3 h) before use. Wood samples were weighed in tin capsules and combusted in an elemental analyser that transfers only the $CO_2$ in helium to the AGE. The $CO_2$ was trapped on zeolite while the helium carrier gas was removed. The $CO_2$ was thermally released and transferred to the reactors by gas expansion. The amount of $CO_2$ was held constant to maintain stable $CO_2/H_2/Fe$ ratios for graphitization at 580 °C (0.9 mg carbon, 4.2 mg iron, $H_2/CO_2$ ratio = 2.3). Water formed from the reduction was frozen in a Peltier-cooled trap (at about –5 °C). The reaction was stopped automatically when residual gas pressures stabilize.

The $^{14}C$ measurements were made with a MICADAS tandem accelerator[18], which detect atoms of specific elements according to their atomic weights[38]. The graphite samples are first bombed with Caesium ions, producing negatively ionized carbon atoms before reaching the tandem accelerator where they are accelerated to the positive terminal by a high voltage difference. At this stage, other negatively charged atoms are unstable and cannot reach the detector. The negatively charged carbon atoms, however, move on to the stripper (a gas or a metal foil) where they lose the electrons and emerge as positively charged carbon atoms. Then, they further accelerate away from the positive terminal and pass through a set of focusing devices where mass analysis occurs. If the charged particles have the same velocity

but different masses, as in the case of the carbon isotopes, the heavier particles are deflected least. Detectors at different angles of deflection then count the particles. All relevant processing steps are summarized in Supplementary Table 3.

**Radiocarbon modelling**. The carbon box model developed and introduced by[12] was herein expanded and improved by separating the NH and SH data to simulate the interhemispheric offset as revealed by the COSMIC tree-ring network (Fig. 3). In the resulting model, each hemisphere contains 11 boxes (Supplementary Fig. 3; Supplementary Table 4), and the carbon exchange between the NH and SH only takes place in the atmosphere and the ocean. Furthermore, the different distribution of land and ocean, the varying growth seasonality and seasonal variation across the tropopause were considered[39]. The carbon fluxes between the boxes are balanced and were adapted using bomb peak data[40]. The model can be fitted simultaneously to two datasets using a $X^2$-fitting routine which evaluates the most likely event date and the additional production needed to reproduce the data (Supplementary Table 4).

**Data uncertainty**. While the above sections, together with the Supplementary Note 1, outline much of the complexity our COSMIC project was facing in terms of data archiving, identification, preparation and processing, further challenges associated with species-specific tree physiological processes of carbon storage and re-mobilization[41], as well as slight differences in growing season lengths[42], and climate-induced intra-annual changes in xylogenesis and carbon allocation[43] could not be entirely neglected. In addition, intra-annual density fluctuations, so-called false rings[44], in semi-arid environments[44], not only require particular care and time for accurate cross-dating, i.e. matching variations in ring width or other tree-ring parameters and characteristics among several, independent tree-ring measurement series from multiple individual trees to allow the identification of the exact calendar year in which each tree ring was formed[45], but may also affect the precise recording of rapid $^{14}C$ increases.

A reduced amplitude, possibly due to species- and site-specific biological carry-over effects, was observed in a Mountain hemlock (*Tsuga mertensiana*) sample from the Great Nunatak Mountain region near the Prince William Sound in Alaska (USA10). A similarly dampened signal was exhibited by a Douglas-fir (*Pseudotsuga menziesii*) sample from the El Malpais National Monument in New Mexico (USA07). Furthermore, repeated measurements along different radii of an individual Scots pine (*Pinus sylvestris*) sample from the Torneträsk region in northern Sweden suggested a one-year inconsistency. The Torneträsk $^{14}C$ measurements herein used, however, precisely detected the cosmic signal in either 993 or 774 CE (depending on the material used). A similar, 1-year dating irregularity was found in an individual kauri (*Agathis australis*) from northern New Zealand. The kauri samples herein used, however, confirm both of the events in 993 and 774 CE. It is important to note at this stage that in the temperate regions of the SH, tree growth starts in the spring of a certain year and ends in the summer of the following year. Following Schulman's (1956) convention[46], which assigns to each tree ring the date of the year in which tree growth started, year 774 in our paper refers to the spring-summer of the NH and November 774 until March 775 on the SH.

On top of biological carry-over effects and wood anatomical features, several biotic and abiotic disturbance factors, such as insect outbreaks, fungal diseases and severe climate anomalies, cannot be ignored, especially if datasets are less replicated. Due to the frequent occurrence of severe defoliation patterns following massive outbreaks of the pandora moth (*Coloradia pandora* Blake), our ponderosa pine (*Pinus ponderosa* Dougl. ex Laws.) sample from a lava flow in central Oregon (USA16) indicated one missing ring. At that time, the sample was not part of a fully crossdated and published chronology. In addition, we detected a missing ring in a dead Bosnian pine (*Pinus heldreichii*) from central Albania (ALB01), as well as in a subfossil Norway spruce (*Picea abies*) remain from the Italian Alps (ITA09). Again, both samples were previously not included in published chronologies. It should be noted in this regard, that the earliest portion of species-specific site chronologies, often truncated and excluded from publications, may rely on the ring width measurements of one or two individual trees only. During such periods of very low sample size, cross-dating cannot be entirely validated. Even if well-replicated composite ring width chronologies are precisely dated at annual resolution, individual measurements of single trees therefore may still sporadically deviate due to missing rings and other reasons described above.

Finally, we suggest that the relatively small amplitude of the $^{14}C$ event in 993 CE, especially when considering limited accuracy in radiocarbon measurements from less precise AMS systems, together with the possible effects of biological memory and even sample contamination in the case of very narrow tree rings and their irregular boundaries, appears less suitable for independent proxy archive dating, especially when compared to the much stronger event in 774 CE.

## Data availability
All data will be freely available via http://www.ams.ethz.ch/research/published-data.html.

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

## Acknowledgements

We thank everyone who participated in fieldwork, sample preparation, cross-dating and/or chronology development. This study was funded by the WSL-internal COSMIC project (5233.00148.001.01), the ETHZ (Laboratory of Ion Beam Physics), the Swiss National Science Foundation (SNF Grant 200021L_157187/1), and as the Czech Republic Grant Agency project no. 17-22102s.

## Author contributions

U.B. and L.W. designed the study with input from J.D.G. Radiocarbon measurements and analyses were performed by J.D.G. and L.W., with the involvement of J.W. Additional radiocarbon measurements and analyses were performed by A.J.T.J., I.P. and F.M. S.A., M.Chr. and L.W. modelled the 14-C. M.B. compiled the historical documentary evidence. The paper was written by U.B., together with P.J.K., C.O., J.E., M.B., F.C.L. and L.W. Further editorial contributions were obtained from J.B., P.C., M.Chr., R.H., F.H., F.H.S., H.-A.S., J.H.S. and R.W. Wood samples were collected, prepared and provided by D.A., M.B, N.B., G.B., A.B., M.Car., D.A.C., P.W.C., E.R.C., R.D'A., N.D., Ó.E., J.E., A.M.F., Z.G., F.G., J.G., H.G.-M., H.G., B.E.G., R.H., F.H., A.H., K.-U.H., V.K., A.K., T.Kol., P.J.K., T.Kyn., A.L., C.L.Q., H.W.L., N.J.L., B.L., V.S.M., K.N., J.P., N.P., M.R., A.S., O.C.(S.), J.H.S., W.T., K.T., R.V., G.W., L.J.W., B.Y. and G.H.F.Y. Ice core data were provided and discussed by M.S.

## Additional information

**Competing interests:** The authors declare no competing interests.

Ulf Büntgen[1,2,3,4], Lukas Wacker[5], J. Diego Galván[2], Stephanie Arnold[5], Dominique Arseneault [6], Michael Baillie[7], Jürg Beer[8], Mauro Bernabei[9], Niels Bleicher[10], Gretel Boswijk [11], Achim Bräuning[12], Marco Carrer [13], Fredrik Charpentier Ljungqvist [1,14,15], Paolo Cherubini[2], Marcus Christl [5], Duncan A. Christie[16,17], Peter W. Clark[18], Edward R. Cook[19], Rosanne D'Arrigo[19], Nicole Davi[19,20], Ólafur Eggertsson[21], Jan Esper[22], Anthony M. Fowler[11], Ze'ev Gedalof [23], Fabio Gennaretti[24], Jussi Grießinger[12], Henri Grissino-Mayer[25], Håkan Grudd [26], Björn E. Gunnarson[15,27], Rashit Hantemirov[28], Franz Herzig[29], Amy Hessl [30], Karl-Uwe Heussner[31], A.J.Timothy Jull[32,33,34], Vladimir Kukarskih [28], Alexander Kirdyanov[1,35,36], Tomáš Kolář[3,37], Paul J. Krusic[1,27,38], Tomáš Kyncl[3], Antonio Lara[16,17], Carlos LeQuesne[16], Hans W. Linderholm [39], Neil J. Loader [40], Brian Luckman[41], Fusa Miyake[42], Vladimir S. Myglan[36], Kurt Nicolussi [43], Clive Oppenheimer [1], Jonathan Palmer [44], Irina Panyushkina [45], Neil Pederson[46], Michal Rybníček[3,37], Fritz H. Schweingruber[2], Andrea Seim[47], Michael Sigl[48], Olga Churakova (Sidorova) [36,49], James H. Speer[50], Hans-Arno Synal[5], Willy Tegel[47,51], Kerstin Treydte[2], Ricardo Villalba[52], Greg Wiles[53], Rob Wilson[19,54], Lawrence J. Winship [55], Jan Wunder[2,11], Bao Yang[56] & Giles H.F. Young[40]

[1]Department of Geography, University of Cambridge, Cambridge CB2 3EN, UK. [2]Swiss Federal Research Institute WSL, CH-8903 Birmensdorf, Switzerland. [3]Global Change Research Institute CAS, 603 00 Brno, Czech Republic. [4]Department of Geography, Masaryk University, 611 37 Brno, Czech Republic. [5]Laboratory for Ion Beam Physics, ETH Zürich, CH-8093 Zurich, Switzerland. [6]Département de biologie, chimie et géographie, University of Québec in Rimouski, QC G5L 3A1, Canada. [7]School of Natural and Built Environment, Queen's University, Belfast BT7 1NN Northern Ireland, UK. [8]Swiss Federal Institute of Aquatic Science and Technology Eawag, CH-8600 Dübendorf, Switzerland. [9]CNR-IVALSA, Trees and Timber Institute, 38010 San Michele all'Adige, TN, Italy. [10]Competence Center for Underwater Archaeology and Dendrochronology, Office for Urbanism, City of Zurich, 8008 Zürich, Switzerland. [11]School of Environment, University of Auckland, 1010 Auckland, New Zealand. [12]Institute of Geography, Friedrich-Alexander-University Erlangen-Nürnberg (FAU), 91058 Erlangen, Germany. [13]Department Territorio e Sistemi Agro-Forestali, University of Padova, 35020 Legnaro (PD), Italy. [14]Department of History, Stockholm University, SE-10691 Stockholm, Sweden. [15]Bolin Centre for Climate Research, Stockholm University, SE-10691 Stockholm, Sweden. [16]Laboratorio de Dendrocronología y Cambio Global, Universidad Austral de Chile, Casilla 567, Valdivia, Chile. [17]Center for Climate and Resilience Research, Blanco Encalada 2002, 8370449 Santiago, Chile. [18]Rubenstein School of Environment and Natural Resources, University of Vermont, Burlington, Vermont 05405, USA. [19]Tree-Ring Laboratory, Lamont-Doherty Earth Observatory of Columbia University, Palisades, NY 10964-8000, USA. [20]Department of Environmental Science, William Paterson University, Wayne, NJ 07470, USA. [21]Icelandic Forest Research Mógilsá, 116 Reykjavik, Iceland. [22]Department of Geography, Johannes Gutenberg University, 55099 Mainz, Germany. [23]Department of Geography, University of Guelph, ON N1G 2W1, Canada. [24]AgroParisTech, INRA, Université de Lorraine, 54000 Nancy, France. [25]Department of Geography, University of Tennessee, Knoxville, TN 37996-0925, USA. [26]Swedish Polar Research Secretariat, SE-104 05, Stockholm, Sweden. [27]Department of Physical Geography, Stockholm University, SE-106 91 Stockholm, Sweden. [28]Institute of Plant and Animal Ecology, Ural Branch of the Russian Academy of Sciences, Ekaterinburg 620144, Russia. [29]Bavarian State Office for Monument Protection, 80539 München, Germany. [30]Department of Geology and Geography, West Virginia University, WV 26505-6300, USA. [31]German Archaeological Institute, 14195 Berlin, Germany. [32]Department of Geosciences, University of Arizona, Tucson, AZ 85721, USA. [33]AMS Laboratory, University of Arizona, Tucson, AZ 85721, USA. [34]Isotope Climatology and Environmental Research Centre, Institute of Nuclear Research, H-4001 Debrecen, Hungary. [35]Sukachev Institute of Forest SB RAS, 660036 Krasnoyarsk, Russia. [36]Department of Humanities, Siberian Federal University, 660041 Krasnoyarsk, Russia. [37]Department of Wood Science, Mendel University in Brno, 61300 Brno, Czech Republic. [38]Navarino Environmental Observatory, GR-24001 Messinia, Greece. [39]Department of Earth Sciences, University of Gothenburg, 405 30 Gothenburg, Sweden. [40]Department of Geography, Swansea University, Swansea SA2 8PP Wales, UK. [41]Department of Geography, University of Western Ontario, London, ON N6A 3K7, Canada. [42]Institute for Space-Earth Environmental Research, Nagoya University, Nagoya 464-8601, Japan. [43]Institute of Geography, University of Innsbruck, 6020 Innsbruck, Austria. [44]Palaeontology, Geobiology and Earth Archives Research Centre, and ARC Centre of Excellence for Australian Biodiversity and Heritage, School of Biological, Earth and Environmental Sciences, The University of New South Wales, Sydney, NSW 2052, Australia. [45]Laboratory of Tree-Ring Research, University of Arizona, Tucson, AZ 85721, USA. [46]Harvard Forest, Harvard University, Petersham, MA 01366, USA. [47]Chair of Forest Growth and Dendroecology, Institute of Forest Sciences, University of Freiburg, Freiburg, Germany. [48]Laboratory of Environmental Chemistry, Paul Scherrer Institute, 5232 Villigen, Switzerland. [49]Institute for Environmental Sciences, University of Geneva, 1205 Geneva, Switzerland. [50]Department of Earth and Environmental Systems, Indiana State University, Terre Haute, IN 47809, USA. [51]Archaeological Service Kanton Thurgau (AATG), 8510 Frauenfeld, Switzerland. [52]Instituto Argentino de Nivología, Glaciología y Ciencias Ambientales, IANIGLA - CONICET, Mendoza, CP 330 5500, Argentina. [53]Department of of Earth Sciences, The College of Wooster, OH 44691, USA. [54]School of Geography and Geosciences, University of St Andrews, St Andrews KY16 9AJ Scotland, UK. [55]School of Natural Science, Hampshire College, Amherst, MA 01002, USA. [56]Key Laboratory of Desert and Desertification, Northwest Institute of Eco-Environment and Resources, Chinese Academy of Sciences, 730000 Lanzhou, China

