## [Peer Review File · Nature Communications]

Reviewers' comments:

Reviewer #1 (Remarks to the Author):

Buntgen et al. have compiled a mammoth data set of radiocarbon measurements over the two rapid rises in radiocarbon activity (often called "Miyake Events") in the late first millennium CE. This rich resource allows them to perform very high resolution analyses of the events, to verify a multitude of dendrochronologies across 5 continents, and to position the anomalies as the pre-eminent chrono-stratigraphic markers for palaeoenvironmental and archaeological research during this time period. In my opinion, the work will attract a wide readership and have a substantial impact on many future studies.

In saying that, I do have a number of comparatively minor scientific, grammatical and typological points to put to the authors, which I will now list in turn.

Line 150 (onward). I understand that the season of the year in which the events occurred is essentially just a modelled estimate, but I have some misgivings about how this result was achieved. There was little comment made throughout the paper about the potential difference in ^{14}C concentration between early and latewood. Apparently all of the rings analysed were treated as amalgams of one year's growth. This begs the question of how sub-annual and ostensibly sub-seasonal precision was obtained. Presumably this had to do with how the global carbon cycle model was configured, and the timing of the growth seasons in the temperate Northern and Southern Hemispheres. However, is such calendrical precision justified, considering the uncertainty in estimating growing seasons one millennium ago, uncertainty in the rapidity with which carbon is directed into tree-rings, and the reuse by some species of stored carbohydrates?

Line 169. The interhemispheric offset is a complex matter; however, I was under the impression that the latest thinking was that it decreased during periods of extreme cold (e.g. Younger Dryas). The claim that lower temperatures in the 770s may be responsible for the opposite effect is a surprise to me, and this claim is not substantiated by any literature reference.

Line 202 onward. I appreciate that it is easy to criticise the validity of the historical literature over these events, but in my opinion it is striking how few relevant records are available. I find the quote about Hensho Knono wholly unconvincing, and the Indian and Mayan evidence tenuous at best, especially given the Maya chronology is by no means fixed. This really just leaves the previously cited (and ambiguous) Anglo Saxon chronicle entry and the debatable translations of Thomas Short. I do not think this part of the paper needs to be emphasised so much. The data stands on its own merits. To reiterate, I am more surprised that chronological archives like the Irish Annals or the dynastic astronomical records from East Asia offer such little support, assuming these were indeed gigantic solar storms.

Line 109. They must be more than just "high-energy solar particles". If the preference is not to refer to the anomalies as "events" then it should at least be implied that extreme fluxes of high-energy solar particles occurred.

Line 113. I wonder if "paragon" is the word sought here, not "paradigm"

Line 117. "volunteer effort" should be "voluntary collaboration" or "act of voluntary cooperation".

Line 126 (and throughout). The passive voice is overused in the article and this sentence is the most discombobulating example. It would be easier to start "Modern compact accelerator mass spectrometry requires less....This method was applied" etc.

Line 146. "times" should be "places".

Line 148. "consisting" should be "consistent".

Line 194. The "Bomb Peak" is usually just referred to in singular.

Line 182. "Independently" should be "Independent".

Reviewer #2 (Remarks to the Author):

Review of Büntgen et al. 'Global signature of cosmic events'

This paper opens a new dimension in the alliance of tree-ring research and the study of cosmogenic isotopes. Already when Miyake et. al in Nature 2012, and 2013 in Nature Communications, presented the quite unexpected level of solar variability documented in radiocarbon (^{14}C) in Japanese tree rings, the potential of this strong marker to synchronize, on a global level, regionally established chronologies became apparent, but so far only few applications were published.

The study of Büntgen and colleagues demonstrates the universal range of the new technique, and for the first time, offers a rigorous validation of 44 tree-ring chronologies from five continents. The principal authors accomplished an impressive task to (1) create a network of dendro-chronologists to collect annually resolved wood samples for two solar events, (2) to perform the 484 ^{14}C analyses in just one AMS laboratory to maintain highest precision and accuracy, (3) model the global ^{14}C re-distribution following the ^{14}C production spikes and (4) evaluate the results in terms of hemispheric and inter-hemispheric mixing in the atmosphere.

So far the search for similar solar events in the past returned just three more candidates (a positive side with respect to the threat to the world), but this publication provides strong arguments to extend annually resolved, tree-ring based ^{14}C data sets for the full range of available tree-ring chronologies.

I strongly recommend this paper for publication in Nature Communications, as it stands.

Reviewer #1

Büntgen et al. have compiled a mammoth data set of radiocarbon measurements over the two rapid rises in radiocarbon activity (often called "Miyake Events") in the late first millennium CE. This rich resource allows them to perform very high-resolution analyses of the events, to verify a multitude of dendrochronologies across 5 continents, and to position the anomalies as the pre-eminent chrono-stratigraphic markers for paleoenvironmental and archaeological research during this time period. In my opinion, the work will attract a wide readership and have a substantial impact on many future studies.

Many thanks for this positive and encouraging summary. We are pleased that our study appears relevant and useful for a broad, international and interdisciplinary audience.

In saying that, I do have a number of comparatively minor scientific, grammatical and typological points to put to the authors, which I will now list in turn.

We carefully considered all comments and suggestions and improved the manuscript accordingly.

Line 150 (onward). I understand that the season of the year in which the events occurred is essentially just a modelled estimate, but I have some misgivings about how this result was achieved. There was little comment made throughout the paper about the potential difference in ^{14}C concentration between early and latewood. Apparently, all of the rings analysed were

treated as amalgams of one year's growth. This begs the question of how sub-annual and ostensibly sub-seasonal precision was obtained. Presumably this had to do with how the global carbon cycle model was configured, and the timing of the growth seasons in the temperate Northern and Southern Hemispheres. However, is such calendrical precision justified, considering the uncertainty in estimating growing seasons one millennium ago, uncertainty in the rapidity with which carbon is directed into tree-rings, and the reuse by some species of stored carbohydrates?

We now better emphasize that the model-based, seasonal dating precision is associated with some uncertainties. At the same time, we better describe the importance of having high-resolution data from both hemispheres to pinpoint the exact intra-annual timing of the two cosmic events. Moreover, we added further details to improve understanding of the model: "This behaviour is indicative of a late boreal spring event. Although associated with uncertainties, our monthly resolved model assumes that about $\frac{2}{3}$ of the total annual wood biomass is produced within 1–2 months in the middle of the vegetation period (Fig. S3). Since this rather short window of cell formation and cell wall thickening is synchronized between all sites in each hemisphere, phenological changes in growing season length have no influence on the model outcome. Moreover, only a very small impact is expected from so-called carry-over effects, because less than 10% of stored carbohydrates from previous year(s) is typically used for cell growth. The model also accounts for seasonal differences in stratosphere-troposphere ^{14}C exchange, and thus simulates the occurrence of both radiocarbon enrichment spikes in the boreal summer (July ± 1 month) of 774 and the boreal spring (April ± 2 months) of 993 (Fig. S4) with great confidence." The methods part has been changed accordingly: "Wood samples, ideally representing the interior part of the ring to avoid boundary contamination and seasonal differences in wood formation, were transported to the Laboratory of Ion Beam Physics at ETH Zurich (Switzerland), where all radiocarbon measurements were performed."

Line 169. The interhemispheric offset is a complex matter; however, I was under the impression that the latest thinking was that it decreased during periods of extreme cold (e.g. Younger Dryas). The claim that lower temperatures in the 770s may be responsible for the opposite effect is a surprise to me, and this claim is not substantiated by any literature reference.

We re-wrote the section and now state that: a) the offset is statistically insignificant, b) any possible climate-induced changes in the inter-hemispheric ^{14}C offset remain debatable, and c) although speculative, these sentences hopefully stimulate discussion and encourage research. The new text reads "Following the LALIA, the boreal summer of 774 over most of the NH was $\sim 0.7^\circ\text{C}$ colder than the 1961–90 reference period²⁰, whereas 993 coincided with medieval summer warmth of $\sim 0.6^\circ\text{C}$ (relative to the 1961–90 mean climatology). Although speculative, slightly lower mean temperatures in the 770s (compared to the 990s) may have contributed to a larger, though insignificant ($4.0\text{‰} \pm 0.4$ in 774 versus $3.5\text{‰} \pm 0.7$ in 993), hemispheric offset in the radiocarbon concentration (Fig. 2). In contrast to previous, model-based assumptions¹⁹, we hypothesize that an overall warmer climate in the 990s might explain the smaller inter-hemispheric ^{14}C difference during high medieval times via reduced atmospheric mixing and/or ocean upwelling in the SH. Our limited understanding of how, it at all, climate change affects the amount of outgassing ^{14}C -depleted CO_2 from the

proportionally larger SH oceans, however, calls for more research into this field. Moreover, different ecological site conditions and species-specific plant physiological processes, including xylogenesis, may further impact the timing and intensity of cellulose-based ¹⁴C anomalies through varying rates of cell formation and carbon sequestration¹².”.

Line 202 onward. I appreciate that it is easy to criticise the validity of the historical literature over these events, but in my opinion, it is striking how few relevant records are available. I find the quote about Hensho Knono wholly unconvincing, and the Indian and Mayan evidence tenuous at best, especially given the Maya chronology is by no means fixed. This really just leaves the previously cited (and ambiguous) Anglo Saxon chronicle entry and the debatable translations of Thomas Short. I do not think this part of the paper needs to be emphasised so much. The data stands on its own merits. To reiterate, I am more surprised that chronological archives like the Irish Annals or the dynastic astronomical records from East Asia offer such little support, assuming these were indeed gigantic solar storms.

We agree and de-emphasized the weight of this section. Moreover, we added more critical wording to stress the limited quality and quantity of the medieval text sources. Changes include: “Given the exceptional nature of these two SEP events, we have examined contemporary medieval texts to see if any references might attest to these cosmic events. We recognise that interpretation of such documents is contested and acknowledge the potential for misinterpreting these narratives.”, as well as “Some references have been interpreted as aurora^{28–30}, but may, alternatively, suggest point-of-light events, such as a gamma ray burst or supernova. Our data, however, suggest globally homogeneous impacts in 774 and 993 that can be best explained by large energy releases from the Sun⁷, such as SEP events. Historical records from Germany, Ireland and the Korean Peninsula suggest the occurrence of red auroras between late-992 and early-993 CE³¹, which could be interpreted as great magnetic storms from intense solar activity. Preceding previously reported ¹⁴C results from local analyses by one year^{5,14}, the medieval observations are consistent with our findings.”.

Line 109. They must be more than just "high-energy solar particles". If the preference is not to refer to the anomalies as "events" then it should at least be implied that extreme fluxes of high-energy solar particles occurred.

The two main sentences of this section have been slight changed to “Often attributed to extreme fluxes of high-energy solar particles^{7,8}, distinct ¹⁴C anomalies in 774/5 and 992-4 CE^{4,5,9–13}, as well as possibly much earlier in 660 and 3372/1 BCE^{14,15} have been identified in local proxy archives. These so-called “Miyake-events” also yield anomalies in records of other cosmogenic radionuclides, such as ¹⁰Be and ³⁶Cl that are measured in ice cores⁷.”.

Line 113. I wonder if "paragon" is the word sought here, not "paradigm".
Corrected.

Line 117. "volunteer effort" should be "voluntary collaboration" or "act of voluntary cooperation".
Changed to " voluntary collaboration of ".

Line 126 (and throughout). The passive voice is overused in the article and this sentence is the most discombobulating example. It would be easier to start "Modern compact accelerator mass spectrometry requires less.... This method was applied" etc.

The section has been rewritten to "Modern compact tandem accelerator mass spectrometry (AMS) requires 1000-times less material and operates as precisely as some traditional decay counters¹⁸ (Methods). Developed and based at ETH-Zurich, the "Mini Radiocarbon Dating System" (MICADAS)¹⁸ was used to measure the ¹⁴C content of 30–50 mg bulk cellulose for a total of 484 tree rings (Methods). A subset of 374 rings in the 770s CE interval originates from 27 records on the Northern Hemisphere (NH) and seven records on the Southern Hemisphere (SH). Another 110 rings that did not reach back into the 8th century CE represent eight NH and two SH records in the 990s CE." Please further note that the utilization of passive voice has been reduced throughout the entire manuscript.

Line 146. "times" should be "places".

Corrected.

Line 148. "consisting" should be "consistent".

Re-written to "Importantly, this seasonal timing is consistent with the observed ~10% relative difference in radiocarbon amplitude between the NH and SH."

Line 194. The "Bomb Peak" is usually just referred to in singular.

Corrected.

Line 182. "Independently" should be "Independent".

Corrected.

Reviewer #2:

Review of Büntgen et al. 'Global signature of cosmic events'

This paper opens a new dimension in the alliance of tree-ring research and the study of cosmogenic isotopes. Already when Miyake et. al in Nature 2012, and 2013 in Nature Communications, presented the quite unexpected level of solar variability documented in radiocarbon (¹⁴C) in Japanese tree rings, the potential of this strong marker to synchronize, on a global level, regionally established chronologies became apparent, but so far only few applications were published.

Many thanks for this kind summary.

The study of Büntgen and colleagues demonstrates the universal range of the new technique, and for the first time, offers a rigorous validation of 44 tree-ring chronologies from five continents. The principal authors accomplished an impressive task to (1) create a network of dendro-chronologists to collect annually resolved wood samples for two solar events, (2) to perform the 484 ¹⁴C analyses in just one AMS laboratory to maintain highest precision and accuracy, (3) model the global ¹⁴C re-distribution following the ¹⁴C production spikes, and (4) evaluate the results in terms of hemispheric and inter-hemispheric mixing in

the atmosphere. So far, the search for similar solar events in the past returned just three more candidates (a positive side with respect to the threat to the world), but this publication provides strong arguments to extend annually resolved, tree-ring based ^{14}C data sets for the full range of available tree-ring chronologies.

We fully agree with this perspective and hope to be able to contribute in an efficient and collegial way to the further improvement of the global radiocarbon record via highest precision, annual ^{14}C measurements for most, if not all of the Holocene (and ideally also beyond).

I strongly recommend this paper for publication in Nature Communications, as it stands.
Many thanks.

REVIEWERS' COMMENTS:

Reviewer #1 (Remarks to the Author):

I thank the authors for addressing the queries I raised in my first review of this manuscript. I understand and agree with the extra information and clarifications they have added. I have no further comments.